# COVID-19 and the Pancreas: A Narrative Review

**DOI:** 10.3390/life12091292

**Published:** 2022-08-23

**Authors:** Emanuele Sinagra, Endrit Shahini, Federica Crispino, Ina Macaione, Valentina Guarnotta, Marta Marasà, Sergio Testai, Socrate Pallio, Domenico Albano, Antonio Facciorusso, Marcello Maida

**Affiliations:** 1Gastroenterology and Endoscopy Unit, Fondazione Istituto G. Giglio, Contrada Pietra Pollastra Pisciotto, 90015 Cefalù, Italy; 2Division of Gastroenterology, National Institute of Research “Saverio De Bellis”, Via Turi, 27, Castellana Grotte, 70013 Bari, Italy; 3Department of Health Promotion Sciences Maternal and Infant Care, Section of Gastroenterology & Hepatology, Internal Medicine and Medical Specialties, PROMISE, University of Palermo, 90133 Palermo, Italy; 4Surgery Unit, Fondazione Istituto G. Giglio, Contrada Pietra Pollastra Pisciotto, 90015 Cefalù, Italy; 5Section of Endocrinology, Department of Health Promotion Sciences, Maternal-Infant Care, Internal Medicine and Specialties of Excellence “G.D’Alessandro” (PROMISE), University Hospital P. Giaccone, University of Palermo, Piazza Delle Cliniche 2, 90133 Palermo, Italy; 6Radiology Unit, Fondazione Istituto G. Giglio, Contrada Pietra Pollastra Pisciotto, 90015 Cefalù, Italy; 7Endoscopy Unit, Department of Clinical and Experimental Medicine, University of Messina, AOUP Policlinico G. Martino, 98122 Messina, Italy; 8IRCCS Istituto Ortopedico Galeazzi, Via Riccardo Galeazzi 4, 20161 Milan, Italy; 9Gastroenterology Unit, Department of Medical Sciences, University of Foggia, 71122 Foggia, Italy; 10Gastroenterology and Endoscopy Unit, S. Elia-Raimondi Hospital, 93100 Caltanissetta, Italy

**Keywords:** COVID-19, SARS-CoV-2, coronavirus, pancreas, pancreatitis

## Abstract

The outbreak of COVID-19, initially developed in China in early December 2019, has rapidly spread to other countries and represents a public health emergency of international concern. COVID-19 has caused great concern about respiratory symptoms, but it is worth noting that it can also affect the gastrointestinal tract. However, the data on pancreatic involvement during SARS-CoV-2 infection are limited. The prevalence and severity of pancreatic damage and acute pancreatitis, as well as its pathophysiology, are still under debate. Moreover, the possible implication of pancreatic damage as an apparent adverse effect of COVID-19 therapies or vaccines are issues that need to be addressed. Finally, the COVID-19 pandemic has generated delays and organizational consequences for pancreatic surgery, an element that represent indirect damage from COVID-19. This narrative review aims to summarize and analyze all the aspects of pancreatic involvement in COVID-19 patients, trying to establish the possible underlying mechanisms and scientific evidence supporting the association between COVID-19 and pancreatic disease.

## 1. Introduction

Severe Acute Respiratory Syndrome Coronavirus 2 (SARS-CoV-2) is responsible for the ongoing pandemic of Coronavirus Disease 2019 (COVID-19) and has caused more than 565 million infections and 6 million deaths worldwide as of 19 July 2022, involving over 200 countries [1].

COVID-19 was initially considered merely as a respiratory disease. Nevertheless, the digestive system can be involved with an incidence that ranges from 3% to 79% [2,3,4]. Pancreatic enzyme elevation and cases of acute pancreatitis (AP) have been attributed to SARS-CoV-2, but the mechanisms of pancreatic damage are still controversial.

In particular, whether the effects on the pancreas are due to the direct tropism of the virus or only an epiphenomenon is still a matter of debate.

The aim of this narrative review is to summarize and analyze the several aspects of pancreatic involvement in COVID-19 patients, as represented in Figure 1.

## 2. Pathogenesis of Pancreatic Damage in COVID-19 Patients

SARS-CoV-2 utilizes angiotensin-converting enzyme 2 (ACE-2) receptors to enter human cells and TMPRSS2 for priming [5]. Such proteins are highly expressed in both gastrointestinal luminal cells [6,7] and pancreatic ductal, acinar and islet cells [8]. According to this, the infection of the gland may be plausible since the virus can spread from the duodenum to the pancreatic duct and then to acinar and islet cells [6], with a cytolytic effect that mediates the releasing of pancreatic amylases and/or lipases [4]. Another explanation for the increase in pancreatic enzymes may be kidney failure. Since the kidneys play a crucial role in the clearance of amylases and lipases from the bloodstream, their malfunction, even if temporary, could lead to their increase [4].

However, it should be considered that the elevation of pancreatic enzymes could be non-specific since it can also be seen in other gastrointestinal diseases (e.g., gastroparesis, gastritis, enteritis and colitis) that have also been reported in COVID-19 patients. Of note, intestinal translocation should be considered [4]. Diarrhea is present in about 2.0–49.5% of COVID-19 patients, and it is known that the viral RNA can be isolated in the stool [4,9,10,11,12]. Therefore, the presence of ACE-2 receptors in the gastrointestinal lumen cells and the consequent alteration of the barrier permeability may result in the reabsorption of the pancreatic enzymes themselves [13,14,15,16]. The occurrence of a cytokine storm is another possible way to induce a pancreatic injury. This condition refers to the pivotal role of the activated immune system as a response to external agents, with a consequent and uncontrolled inflammatory systemic response [17,18,19,20,21,22,23]. In the severe forms of COVID-19, there is a release of a wide spectrum of cytokines, such as interleukin IL-2, IL-6, IL-8, IL-7, interferon-γ and tumor necrosis factor-α [22,23,24] that may cause pancreatic damage. Finally, the drug-induced pancreatic injury should be considered among the possible causes of pancreatic enzyme elevation in COVID-19 patients. Among the drugs that have been reported to cause damage in the pancreas, there are antivirals (such as remdesivir, and lopinavir/ritonavir), antipyretics belonging to non-steroidal anti-inflammatory drugs (NSAIDs) and tocilizumab and baricitinib [4]. Tocilizumab, and baricitinib, have been proposed for the treatment of COVID-19, and have been related with the development of AP, and hypertriglyceridemia, an established etiology of AP. Lopinavir/ritonavir, which are associated with lipid metabolism abnormalities, have not been involved as causative agents of AP. Propofol infusion in critically ill patients increases serum triglyceride levels, secondary to the lipid emulsion vehicle, which can contribute to hypertriglyceridemia and pancreatic injury. Finally, Doxycycline, lisinopril, estrogens and steroids are associated with AP development, and constituted the chronic medications of some of COVID-19 patients.

In Figure 2, the mechanisms involved in pancreatic damage in COVID-19 patients are represented.

## 3. Clinical Evidence of Pancreatic Enzyme Elevations and Acute Pancreatitis in COVID-19 Patients

To date, animal models of COVID-19 that develop pancreatic damage are still lacking. With regard to clinical evidence on humans, several case reports/series, retrospective, case-control and cohort studies have detailed a connection between pancreatic enzyme elevations, AP and COVID-19, although the prevailing evidence from real-world data is questionable [5]. Elevated amylase or lipase have been reported in 8.5–17.3% of COVID-19 positive patients, while AP defined by Atlantic criteria is less frequent, in 1.8–2.0% of patients with increased values of these pancreatic enzymes [8,18,25,26].

In particular, it is not easy to answer if COVID-19 could be thought of as a new etiology of AP. Usually, viral-attributed AP is infrequent, and it has been found more frequently only in immunocompromised patients. Furthermore, some considerations cannot be excluded from this review.

It should be noted that some studies define AP only by an increase in amylase and lipase, without meeting the Atlanta criteria, while others have determined the pancreatic injury definition differently [e.g., pancreatic abnormality has been described as a rise of serum amylase and lipase levels above the upper normal limit (UNL)]. Furthermore, imaging findings resulting from computed tomography (CT) and magnetic resonance (MRI) anomalies have been described by only some studies. Therefore, conditions such as micro lithiasis or pancreatic cystic lesions could be missed, thus augmenting the rate of cases of unknown origin attributed to COVID-19.

Finally, the temporality of SARS-CoV-2 infection and the development of AP is highly different and generally underrated in the studies: some patients develop COVID-19 symptoms and abdominal pain when the infection begins, whereas others present with AP several days after COVID-19 diagnosis. 

In conclusion, as some authors suggest, a multiple-hit theory in AP has been hypothesized since multiple etiological factors can be involved in AP development: viral direct damage, thrombogenic state of COVID-19 and certain drugs, as aforementioned.

Evidence regarding pancreatic damage in COVID-19 patients are showed in Table 1.

## 4. Case Reports/Series of Acute Pancreatitis in Patients with COVID-19

One of the initial reports was the AP development in two over three COVID-19 positive hospitalized members of a Danish family [27]. Later on, Elhence et al. described three cases of severe AP with respiratory failure testing positive for COVID-19 several days later than the diagnosis of AP, and three cases of hospitalized AP developing hospital-acquired COVID-19 [28].

In a cohort of thirty-five patients with AP, Szatmary et al. observed ten cases with SARS-CoV-2 positivity, of whom five had no definite etiology and were attributed to COVID-19 infection [29].

An interesting case has also been described of a patient with autoimmune pancreatitis who developed AP and COVID-19 during a course of prednisone [30].

## 5. Retrospective Studies Regarding Acute Pancreatitis in Patients with COVID-19

A 2020 multicentric Spanish study by Miro et al. involving over 63,000 COVID-19 patients from emergency rooms before hospitalization, recorded a 0.07% frequency of AP whose etiology was not defined by the authors [31].

A 2020 multicentric US study by Inamdar S et al. analyzing 11,883 hospitalized patients with COVID-19, revealed a low prevalence between the concomitant occurrence of SARS-CoV-2 and AP with a 0.43% incidence and 0.27% in SARS-CoV-2-negative patients. Patients with both AP and COVID-19 were more likely to need mechanical ventilation and had a longer length of hospitalization than patients with AP without COVID-19 (OR = 5.65; *p* = 0.01 and OR = 3.22; *p* = 0.009, respectively) [32]. Similarly, another US work by Dirweesh A et al. involving 75 COVID-19 positive patients from 339 patients with AP, showed that the mortality rate was higher in patients with both AP and COVID-19 when compared to negative cases (*p* = 0.004) [33].

In an English study by Stephens JR et al. among 234 COVID-19 patients included, 67.5% (*n* = 158) had an abnormally raised elevation in amylase level during the patient admission to the intensive care unit (ICU), and in 22.2% of cases (*n* = 52) the peak value was above 3 times ULN. However, only 1.7% (*n* = 4) met the revised Atlanta criteria for AP, and among these patients, 50% died (*n* = 2/4) [34]. Similarly, a 2020 Turkish study by Akkus C et al. showed that among 127 COVID-19 patients tested for lipase, 15.7% (*n* = 20) had serum hyperlipasemia above the ULN but only 0.01% (*n* = 2) were diagnosed with AP. Interestingly, in the high lipase group, there was a higher prevalence of diabetes mellitus, a major need for ICU admission and a more prolonged time of hospitalization than the controls (*p* = 0.002; *p* = 0.027 and *p* = 0.003, respectively). The authors gathered that pancreatic injuries or AP may originate during SARS-CoV-2 infection, principally in patients with pre-existing diabetes mellitus, in whom the monitoring of pancreatic enzymes becomes essential due to poorer outcomes [35]. Another multicenter 2021 US study by Ramsey ML et al. recorded 400 patients with hyperlipasemia in 1992 COVID-19 positive patients. However, among them, they found only two patients who presented AP, of which both had biliary etiology, thus concluding that hyperlipasemia was from critical illness as it was associated with the mechanical ventilation requirement (OR, 2.55; 95% CI, 1.6–4.08) and not pancreatitis [36]. Lately, in a 2021 population-based multicenter study by Singh RR et al. [37], 0.32% of COVID-19 patients (*n* = 1406/435,731) had hyperlipasemia which was associated with higher 30-day mortality (OR = 1.53, *p* < 0.001), risk of acute kidney injury (OR = 1.5, *p* = 0.003) and vasopressor use (OR = 1.69, *p* < 0.001) in the absence of a certain diagnosis of AP. On the other hand, in another 2020 US study by McNabb-Baltar J et al. asymptomatic hyperlipasemia in hospitalized COVID-19 patients was not associated with poor outcomes [38].

Regarding the elevation of the amylase level, in 2021 Bacaksiz F et al. [39] reported a 23% hyperamilasemia rate in Turkish people affected by COVID-19 disease (*n* = 316/1378). In such patients, the amylase levels increased one to three times in 18.9% (*n* = 261) and three times in nearly 4% of patients (*n* = 55/1378). In addition, AP was detected in only 1.9% (*n* = 6) following the Atlanta criteria. At the univariate and multivariate analyses, hyperamilasemia was significantly associated with COVID-19 severity (OR = 4.37; *p* < 0.001). Furthermore, diabetes mellitus (OR = 1.82; *p* = 0.001), kidney failure (OR = 5.18; *p* < 0.001), liver damage (OR: 6.63; *p* < 0.001), hypotension (OR = 6.86; *p* < 0.001) and sepsis (OR = 6.20; *p* = 0.008) were correlated with COVID-19-related mortality. 

A 2021 Italian study [40] recorded that 26% of COVID-19 patients (*n* = 66/254) had a mild elevation of pancreatic enzymes, and 4.3% (*n* = 11) had severe elevation (over than 3 times ULN). Overall, only 0.78% (*n* = 2) met the AP diagnostic criteria. The multivariate analysis determined that the pancreatic enzyme elevations were significantly associated with ICU admission (OR = 5.51, *p* < 0.0001). Therefore, the authors assumed that an increase in serum pancreatic enzymes, but not AP, is common in hospitalized COVID-19 patients and is connected with ICU admission.

Additionally, regarding the prognosis of pancreatitis and COVID-19, a study by Karaali R et al. has shown that patients with AP and COVID-19 had a higher need for ICU admission and a higher rate of severe AP than negative ones (7.2% vs. 0.9%; *p* < 0.01, 32.5% vs. 14.1%; *p* < 0.03) [41]. Additionally, COVID-19 positivity and the presence of COVID-19 pneumonia significantly and negatively influenced mortality (*p* < 0.05) [41].

Another 2021 US study by Kumar V et al. enrolled 17 COVID-19-positive patients, among 985AP ones. Nine of them were admitted for respiratory failure and developed AP after a median of 22.5 days from the onset of COVID-19 symptoms, while the other half were admitted for abdominal pain. Patients admitted primarily with severe COVID-19 disease were younger (median age 57 years vs. 63 years), females (55.6% vs. 25.0%), of Hispanic ethnicity (55.6% vs. 25.0%) and obese (88.9% vs. 37.5%) and had worse outcomes than the ones that exhibited AP on admission [42]. 

## 6. Cohort and Case-Control Studies Regarding Acute Pancreatitis in Patients with COVID-19

The 2020 Turkish study by Akarsu C et al. prospectively considered data from 316 patients with COVID-19-pneumonia, of whom 12.6% (*n* = 40/316) were affected by AP. Hospitalization and mortality rates were higher in COVID-19 patients with AP than those without AP (*p* = 0.0038 and *p* < 0.0001, respectively). The authors concluded that the appearance of pancreatic damage can worsen the clinical condition of COVID-19 patients, raising the mortality rates [43].

Another 2020 US study by Barlass U et al. registered that 16.8% of COVID-19 cases (*n* = 14/83) were with elevated lipase levels (over 3-fold UNL). Compared with lower lipase levels (lower than 3-fold UNL), patients with elevated lipase had greater rates of ICU admission (92.9% vs. 32.8%; *p* < 0.001) and intubation (78.6% vs. 23.5%; *p* = 0.002) [26,44]. A significant association between severe COVID-19 and pancreatic injury was also shown in the 2020 Chinese study by Liu et al. [8,45], in which the patients with COVID-19-severe-pneumonia had higher levels of pancreatic enzymes compared to the COVID-19-non-severe ones. These results were in line both with a German study [46] showing that hyperlipasemia is a common finding in Acute Respiratory Distress Syndrome (ARDS) related to COVID-19, and with a US study [47] reporting that hyperlipasemia was associated with higher mortality rates than normal lipase patients (32% vs. 23%, OR = 1.6, 95%CI 1.2–2.1, *p* = 0.002). Furthermore, a 2021 international multicenter study by Pandanaboyana S et al. [48] has shown that patients with AP and SARS-CoV-2 infection were at a higher risk of a severe form of AP and, consecutively, at a higher risk of poor clinical outcomes (OR = 2.77, *p* < 0.003), prolonged length of hospital stay (OR = 1.32, *p* < 0.001) and 30-day mortality (OR = 2.41, *p* < 0.04). On the contrary, a 2021 cross-sectional study [49] carried out on 433 COVID-19 patients proved that only 1.2% (*n* = 5/433) of patients had an uncomplicated form of AP, displaying that COVID-19-related AP is of mild clinical impact.

## 7. Systematic Reviews, Pooled Analyses and Meta-Analyses Regarding Acute Pancreatitis in Patients with COVID-19

A 2021 systematic review on 35 articles, including overall 37 patients, summarized that AP might be the first symptom of COVID-19 [50]. Serum amylase and lipase levels were raised in almost all the patients. In particular, in more than half, their levels exceeded 3-fold ULN, and pancreatic necrosis was reported only in 6% [50].

A 2021 US pooled analysis [51] of patients with COVID-19 included 6 retrospective observational studies and 1 prospective observational study: among 756 COVID-19 patients, the pooled hyperlipasemia prevalence was 11.7% (95%CI, 0.094–0.140; *p* = 0.001; I2 = 0%) and the pooled OR for severe COVID-19 (e.g., ICU admission, mechanical ventilation or death) in patients with hyperlipasemia was 3.14 (95%CI, 1.543–6.400; *p* = 0.003). Accordingly, the authors gathered that severe pancreatic injury resulting in AP might not be a common event in COVID-19 patients, in whom hyperlipasemia holds a 3-fold higher risk of poor clinical outcomes.

Additionally, a 2021 meta-analysis by Mutneja HR et al. [52] included four observational studies: in 2419 patients with AP the presence of COVID-19 significantly increased the chance of mortality (OR = 4.10, 95%CI 2.03–8.29), the incidence of severe AP (OR 3.51, 95%CI 1.19–10.32) and of necrotizing pancreatitis (OR = 1.84, 95%CI 1.19–2.85) than in non-COVID patients [52]. Consequently, the authors proposed that COVID-19 may negatively influence the morbidity and mortality linked with AP.

## 8. Acute Pancreatitis after COVID-19 Vaccine

According to Pfizer’s registration data, two cases of AP were witnessed during phase 2/3 clinical trial of the COVID-19 mRNA vaccine amongst 38,000 American participants [53]. Further, six cases of AP were registered among 40,883 UK reports from databases handling adverse reactions for the mRNA Pfizer/BioNTech vaccine [54]. In detail, Parkash O et al. [55] described a case of a 96-year-old female with a prior history of cholecystectomy who developed an uncomplicated self-limiting idiopathic AP after the first dose of the Pfizer-BioNTech COVID-19 mRNA vaccine. A similar case was reported by Walter T et al. [56] of a 43-year-old European male that developed idiopathic AP after the Pfizer-BioNTech COVID-19 mRNA vaccine. A further case report by Cieślewicz A et al. [57] noticed a CRP and urine amylase increase without any associated morphological pancreatic changes on imaging in a young Caucasian female one day after receiving the first dose of the Pfizer BioNTech COVID-19 mRNA vaccine.

## 9. COVID-19 and Diabetes Mellitus

There is a bidirectional relationship between COVID-19 and diabetes. On the one hand, type 1 (T1DM) and type 2 diabetes mellitus (T2DM) are sharply associated with a significantly raised risk of in-hospital death and of great disease severity during the COVID-19 pandemic, especially in older COVID-19 patients with renal or cardiac disease [58,59,60,61].

On the other hand, new-onset diabetes, and severe metabolic complications of pre-existing diabetes, including diabetic ketoacidosis (DKA) and hyperosmolarity, have been observed in patients with COVID-19. Admittedly, a US multicenter study has shown that, out of 64 adults with T1DM who had established or presumed COVID-19, more than half developed hyperglycemia, and nearly one-third recorded DKA [62]. New-onset hyperglycemia is being increasingly described in adults with COVID-19 without a history of diabetes. For instance, a study conducted on 1122 hospitalized US COVID-19 patients reported that 451 patients had hyperglycemia and, among them, 257 had uncontrolled hyperglycemia [63]. Likewise, a 2020 UK multicenter study revealed an increase in new-onset TDM1 in children with SARS-CoV-2 infection [64].

It is thought that infection-induced inflammation could result in insulin resistance and consequent stress-hyperglycemia [58]. However, it is uncertain to what degree the direct viral destruction of islet cells, via ACE-2, and consequent lowered insulin production and release might contribute. Indeed, new-onset diabetes and exocrine insufficiency are more prevalent in severe cases of AP [65,66]. Evidence regarding occurrence of diabetes mellitus in COVID-19 patients are showed in Table 2.

## 10. Postponement of Pancreatic Surgery during COVID-19 Pandemic

COVID-19 delayed pancreatic cancer (PC) diagnosis, staging and surgery. Different surveys reported that, in 2020, the histologically proven PC diagnostic yield was significantly reduced as compared to 2019 [67]. This is in contrast only to a single Japanese retrospective study that did not show any significant reduction in the newly diagnosed PC during the pandemic emergency [68].

A UK study from 29 pancreatic centers recorded that, during the initial phase of the COVID-19 pandemic, less than 25% had regular availability of diagnostic and staging tools for PC, while 20% were unable to perform surgery [69]. The last data were confirmed by another survey in which more than half of the pancreatic centers recorded less pancreatic surgery, lessening the weekly pancreatic resection rate from 3 (IQR: 2–5) to 1 (IQR: 0–2), *p* < 0.001 [70]. To overcome these organizational difficulties, the need for a tailored treatment emerged as an issue that needs to be assessed. A European survey reported that the majority of pancreatic centers prioritized pancreatic cancer patients according to the disease stage, age and concurrent comorbidities [71]. Moreover, the UK survey cited above displayed that the elderly and those with chronic pulmonary diseases were less likely to be surgically managed throughout the pandemic. Furthermore, one-third of UK centers revised their standard therapeutic pathway from surgery-first to neoadjuvant chemotherapy. In this regard, it is important to highlight that the history of chemotherapy in the previous one month did not show to have any significant effects on COVID-19 mortality when compared with PC patients who did not receive recent chemotherapy [69].

In addition, concretely, Kato H et al. [72] suggested the postponement of both surgery for intraductal papillary mucinous neoplasms (IPMNs) with only worrisome features and for small pancreatic neuroendocrine tumors without lymphadenopathy or clinical symptoms. An analysis of a 2021 US prospective multicenter study of PC surveillance revealed that COVID-19-delayed surveillance in high-risk patients did not cause adverse outcomes in patients efficiently rescheduled. In fact, amongst 693 high-risk surveilled patients, 16% of them had an endoscopic ultrasound (EUS) programmed during the COVID-19 pandemic, with 97% of these EUS being cancelled. Of the cancelled EUS surveillance, 83% were scheduled again in a median of 4.1 months; however, 17% were not scheduled again after 6 months follow-up. Favorably, no PC was diagnosed among those undergoing delayed surveillance [73].

## 11. Postponement of Pancreatic Endoscopy during COVID-19 Pandemic

The initial COVID-19 pandemic spike prompted international recommendations to postpone nonurgent endoscopic procedures [74,75,76]. This was especially critical in the case of pancreatic cancer, because routine pancreatic endoscopic examinations may be restricted, and disease biology precludes significant treatment delays. Notably, endoscopic procedures decreased by 72.9%, and overall cases of pancreatic-biliary cancers decreased by 23.4% in a 2020 multicenter Italian study as endoscopic activity was reduced during the lockdown period [77,78,79]. Another Italian survey conducted in 2021 compared to 2019 found significant differences in pancreatic cancer between 2017 and 2019. In the 2019–2020 comparison, the north (−14.1%) and center (−4.7%) had fewer pancreatic cancers, while the south (+12.3%) had an increase [67,80]. In contrast, a 2021 retrospective Japanese study compared the number of pancreaticobiliary endoscopies and newly diagnosed pancreaticobiliary cancers before, during and after the emergency declaration (1 April 2018 to 6 April 2020, 7 April to 25 May 2020 and 26 May to 31 July, respectively). The monthly prevalence of pancreatic cancer diagnoses as pancreaticobiliary endoscopies for EUS and ERCP did not differ significantly between the three groups (8.0/7.5/7.5 cases, *p* = 0.5; 67.8/62.5/69.0 cases, *p* = 0.7; and 89.8/51.5/86.0 cases, *p* = 0.06). Despite this, the number of EUS cases decreased by 42.7% between before and after the declaration of an emergency [68,81,82].

## 12. Pancreatic Transplantation

Of 2084 UK pancreatic transplants performed in 2020, 3.4% of patients appeared positive to SARS-CoV-2 but only 0.5% died [74].

Additionally, as reported in the US case series by Dube GK et al. [1,45], the COVID-19 transplanted patients presented with symptoms (fever in 100%, cough in 75%), disease severity spectrum and clinical courses comparable to what were revealed in other solid organ transplant patients [75].

## 13. Conclusions

Laboratory abnormalities suggestive of pancreatic damage and AP have been reported in the setting of COVID-19 infection. Among COVID-19 patients, the prevalence of hyperamylasemia ranged from 15.7% to 33% [8,32,35], while the pooled prevalence of hyperlipasemia in patients with COVID-19 was 11.7% [51]. Notably, despite the high prevalence of raised serum amylase and lipase levels in COVID-19 critically ill patients, only a minority had a confirmed diagnosis of AP [34]. The severity and mortality of AP can increase in these patients, especially in the presence of COVID-19-related pneumonia [41,48,52]. The prevalence of an established AP in COVID-19 patients has been documented in 12.6% and 52.9% of patients admitted for COVID-19 respiratory failure [42,43]. Conversely, amongst patients admitted with a diagnosis of AP, the prevalence of those who tested positive for COVID-19 ranged from 8.3% to 23% [29,33,36,48]. As the pandemic progresses, special attention should be given to the evaluation of pancreatic damage, its clinical implication and pancreatic cancer so that faster diagnosis can enable faster implementation of treatment.

Finally, it is necessary to make national or international guidance uniform to guarantee a standard level of care for all patients expecting pancreas surgery and pancreatic transplants during the COVID-19 pandemic.

## Figures and Tables

**Figure 1 life-12-01292-f001:**
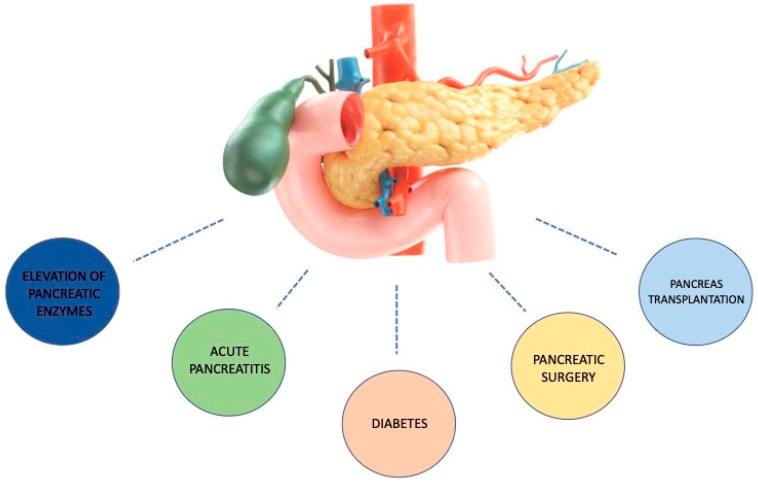
Possible pancreatic involvement in COVID-19.

**Figure 2 life-12-01292-f002:**
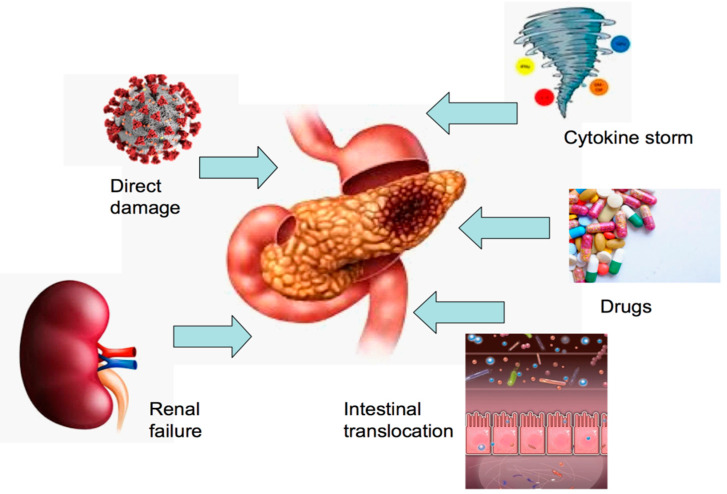
Mechanisms involved in pancreatic damage in COVID-19 patients.

**Table 1 life-12-01292-t001:** Evidence regarding pancreatic damage in COVID-19 patients.

First Author (year)	Region	Study Design	No. of Pts with Pancreatic Injury/Total No. of Patients	Raised Amylase	Raised Lipase	Remarks
Liu [4,8,9,10,11,12,13,14,15,16,17,18,19,20,21,22,23,24,25,26,27,28,29,30,31,32,33,34,35,36,37,38,39,40,41,42,43,44,45] (2020)	China	RS	13/121 (10.74%)	13/121 (10.74%)	12/121 (9.92%)	Non severe cases: 1.85%; Severe: 17.9%; NSAIDS 2; Corticosteroids 4
Wang [18] (2020)	China	CHS	9/52 (4.68%)	7/9 (77.7%)	3/9 (33.3%)	Potential mild pancreatic injury patterns in pts with COVID pneumonia
Bruno [25] (2021)	Italy	CHS	6/70 (8.5%)	6/70 (8.5%)	6/70 (8.5%)	Pancreatic abnormality in one case at admission; associated GI symptoms in all; no definite clinical AP
Barlass [26,27,28,29,30,31,32,33,34,35,36,37,38,39,40,41,42,43,44] (2020)	US	CCS	14/83 (16.8%)	NA	14/83 (16.8%)	Elevated lipase associated with increased ICU admission and intubation
Hadi [27] (2020)	Denmark	CS	2/3 (66.6%)	1/3 (33.3%)	1/3 (33.3%)	The two cases had severe acute pancreatitis with multiorgan failure, but it is not possible to evaluate if the acute pancreatitis contribuited to the severe course of the disease
Liaquat [30] (2020)	US	CR	1	NA	+	During course of prednisone
Miro [31] (2020)	Spain	MS	45/63.822 (0.0007%)	NA	NA	The etiology of AP was not defined
Inamdar [32] (2020)	US	MS	189/11.883 (0.01%)	NA	NA	Pts with pancreatitis who were also COVID-19 positive were more likely to require mechanical ventilation and had longer length of hospital stay compared to pts with pancreatitis without COVID-19
Dirweesh [33] (2020)	US	RS	75/339 (22.1%)	NA	NA	Mortality was significantly higher in pts with AP and coexisting COVID-19, also higher incidence of MOF and POF
Stephens [34] (2021)	UK	RS	158/234 (67.5%)	158/234 (67.5%)	NA	Only a minority of pts with a raised serum amylase concentration had a confirmed diagnosis of AP as defined by the rAC
Akkus [35] (2020)	Turkey	RS	127/309 (15.7%)	NA	20/309 (6.4%)	Pancreatic injuries or AP may develop during COVID-19 infection, especially in those with pre-existing DM
Ramsey [36] (2021)	US	MS	400/1992 (20.08%)	NA	400/1992 (20.08%)	Elevated serum lipase in COVID-19 pts likely reflects the overall disease severuty
Singh [37] (2021)	US	MS	1406/435,731 (0.32%)	NA	1406/435,731 (0.32%)	Worse clinical outcomes
McNabb-Baltar [38] (2020)	US	RS	9/71 (12.1%)	NA	9/71 (12.1%); 2/71 (2.8%) had levels >3 * ULN	Associated GI symptoms; no clinical AP
Bacaksiz [39] (2021)	Turkey	RS	316/1378 (23%)	316/1378 (23%)	NA	Hyperamilasemia was significantly associated with COVID-19 severity
Troncone [40] (2021)	Italy	RS	254/282 (90.07%)	254/282 (90.07%)	254/282 (90.07%)	An increase in serum pancreatic enzymes, but not AP, is common in hospitalized COVID-19 pts and is connected with ICU admission
Karaali [41] (2021)	Turkey	RS	189/562 (33.6%)	NA	NA	AP and COVID-19 had a higher need for ICU admission and higher rate of severe AP
Kumar [42] (2021)	US	RS	17/985 (1.7%)	NA	NA	Pts with COVID-19 who had acute pancreatitis on admission had more benign course and overall better outcomes as compared to pts who developed AP during hospitalization
Akarsa [43] (2020)	Turkey	CCS	40/316 (12.6%)	NA	NA	Mortality rate may increase in these pts
Rash [46] (2021)	Germany	CHS	22/38 (57.8%)	NA	10/38 (26%)	Pts with lipasemia were significantly longer on mechanical ventilation than patients with COVID-19 associated ARDS
Ahmed [47] (2021)	US	CCS	992/5597 (17.7%)	NA	429/5597 (7.6%)	Pts with hyperamylasemia had a more complicated hospitalizationcourse with a higher number of ICU admissions and request of mechanical ventilation
Pandanaboyana [48] (2021)	UK, Turkey	CHS	149/1777 (8.3%)	NA	NA	SARS-CoV-2 infection in acute pancreatitis increases disease severity and 30-day mortality
Bulthius [49] (2021)	Netherlands	CSS	5/433 (1.2%)	15/433 (15%)	15/433 (15%)	COVID-19-related AP is rare and of little clinical impact
Bircakova [50] (2021)	Czech Republic	SR	16/37 (43%)	26/30 (87%)	31/31 (100%)	The association between pancreatic injury and GI symptoms is strong
Goyal [51] (2021)	US	SR	752	NA	92/752 (11.7%)	Severe pancreatic injury resulting in AP might not be a common event in COVID-19
Mutneja [52] (2021)	US	SR	2419	NA	NA	COVID-19 adversely impacts morbidity and mortality associated with AP

Table legend: CS = case series; CR = case reports; RS = retrospective studies; MS = multicentric study; CCS = case-control study; CHS = cohort study; PS = population study; SR = systematic review; CSS = cross-sectional study; NA = not available.

**Table 2 life-12-01292-t002:** Evidence regarding occurrence of diabetes mellitus in COVID-19 patients.

First Author (year)	Region	Study Design	No. of Cases	Remarks
Holman [59] (2020)	UK	CHS	10,525	Death in people with type 1 and type 2 diabetes rose sharply during the initial COVID-19 pandemic
Barron [60] (2020)	UK	PS	263,830 (0·4%) had a recorded diagnosis of type 1 diabetes; 2,864,670 (4·7%) had a diagnosis of type 2 diabetes	Type 1 and type 2 diabetes are both independently associated with significant increased odds of in-hospital death with COVID-19
Gregory [61] (2021)	US	CHS	160 with type 1 diabetes, and 273 with type 2 diabetes	Diabetes status independently increases the adverse impacts of COVID-19
Ebekozien [62] (2020)	US	MS	64	COVID-19 pts developed hyperglycemia and 1/3 DKA
Bode [63] (2020)	US	CHS	1122	Among hospitalized pts with COVID-19, diabetes and/or uncontrolled hyperglycemia occurred frequently
Unsworth [64] (2020)	UK	CHS	30	Apparent increase in new-onset type 1 diabetes in children during COVID-19 pandemic

Table legend: CS = case series; CR = case reports; RS = retrospective studies; MS = multicentric study; CCS = case-control study; CHS = cohort study; PS = population study; SR = systematic review; CSS = cross-sectional study; NA = not available.

## Data Availability

Not applicable.

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
