# Peer review of "COVID-19 and the Pancreas: A Narrative Review"

_life, 2022, doi:10.3390/life12091292_

Round 1

Reviewer 1 Report

This is a well written narrative review of COVID 19 infection and pancreas involvement. I have few suggestions as below

Introduction is well written.

Please revise the manuscript for spell errors and grammatical errors. 

Figure 1: I think it is redundant and recommend to remove this. 

Line 63: Do authors mean non specific?

Line 92: Which lab abnormality is suggestive of pancreatic damage? Elevated lipase and amylase can be seen in other system involvement like hydronephrosis as well. Better to term this as elevated lipase or amylase is seen in ...

Line 110. Please comment on the actual infection vs drug induced pancreatitis vs any immune phenomenon/SIRS

Lines 254: DM is a risk factor for COVID and kindly emphasize on that as you stated in previous statement that the relation is bidirectional. DM increases the risk of death among covid patients is slightly tangential. Kindly rephrase. 

Author Response

Response to reviewers’ comments

Dear Editors, dear Reviewers,

We wish to express our appreciation to the Editors and Reviewers for their insightful comments, which have helped us significantly to improve our manuscript. According to the suggestions, we have thoroughly revised our manuscript and its final version is enclosed. Point-by-point responses to the comments are listed below.

Reviewers’ comments #1

- Introduction is well written

Response: Dear reviewer, many thanks for Your valuable appreciation

  • Please revise the manuscript for spell errors and grammatical errors. 

Response: We edited the text after a review by a native English speaker

  • Line 63 - Do authors mean non specific

Response: Yes, and we corrected the word according to Your valuable suggestion

  • Line 92: Which lab abnormality is suggestive of pancreatic damage? Elevated lipase and amylase can be seen in other system involvement like hydronephrosis as well. Better to term this as elevated lipase or amylase is seen in ...

Response: Dear Reviewer, we added “elevated amylase or lipase is seen” as You suggested

  • Line 110. Please comment on the actual infection vs drug induced pancreatitis vs any immune phenomenon/SIRS

Response: Dear Reviewer, we added the following comment: “In conclusion, as some authors suggest, a multiple-hit theory in AP has been hypothesized since multiple etiological factors can be involved in AP development: viral direct damage, thrombogenic state of COVID-19, and certain drugs, as aforementioned.”

  • Lines 254: DM is a risk factor for COVID and kindly emphasize on that as you stated in previous statement that the relation is bidirectional. DM increases the risk of death among covid patients is slightly tangential. Kindly rephrase. 

Response: Dear Reviewer, we rephrased accordingly. “On the one hand, type 1 (T1DM) and type 2 diabetes mellitus (T2DM) are sharply associated with a significantly raised risk of in-hospital death and of great disease severity during the COVID-19 pandemic, especially in older COVID-19 patients with renal or cardiac disease [59-62].

On the other hand, new-onset diabetes and severe metabolic complications of pre-existing diabetes, including diabetic ketoacidosis (DKA) and hyperosmolarity, have been observed in patients with COVID-19.”

Many thanks again

Sincerely Yours

Emanuele Sinagra

Reviewer 2 Report

I’ve read with great interest the manuscript “COVID-19 and pancreas: a narrative review”, which describes pancreatic involvement during SARS-CoV2 infection, its treatment and consequences on centers dedicated to pancreatic pathology.

The topic of the review is actual and interesting for the readers. Figures are well designed and highly relevant for the information presented.

The manuscript is well structured, regarding the association of AP and COVID-19, COVID-19 vaccine and impact on pancreatic surgery. In the paragraph on pathogenesis, the authors detail that pancreatic involvement during COVID-19 might also occur due to COVID-19 related medication – a dedicated sub-section summarizing literature data on this would be recommended, if possible.  

Also, considering the high prevalence of hyperamylasemia/ hyperlipasemia among COVID-19 patients, but the low prevalence of AP in these patients, the manuscript would increase in value if the authors provided an algorithm/decision tree on how to approach increase in pancreatic serum enzymes in COVID-19 patients. Besides the impact on pancreatic surgery, the authors could discuss the impact on pancreatic endoscopy also.  

There is a small error on page 1, Line 38, “565 infections and 6 million deaths” – missing “million” for the no of infections.

Author Response

Response to reviewers’ comments

Dear Editors, dear Reviewers,

We wish to express our appreciation to the Editors and Reviewers for their insightful comments, which have helped us significantly to improve our manuscript. According to the suggestions, we have thoroughly revised our manuscript and its final version is enclosed. Point-by-point responses to the comments are listed below.

Reviewers’ comments #2

 - The manuscript is well structured, regarding the association of AP and COVID-19, COVID-19 vaccine and impact on pancreatic surgery. In the paragraph on pathogenesis, the authors detail that pancreatic involvement during COVID-19 might also occur due to COVID-19 related medication – a dedicated sub-section summarizing literature data on this would be recommended, if possible. 

Response: Dear reviewer, we added a brief detailed review about this topic: Tocilizumab, and baricitinib, has been proposed for the treatment of COVID-19, has been related with the development of AP, and hypertriglyceridemia, an established etiology of AP.  Lopinavir/ritonavir, which are associated with lipid metabolism abnormalities, have not been involved as causative agents of AP. Propofol infusion in critically ill patients increases serum triglyceride levels, secondary to the lipid emulsion vehicle, which can contribute to hypertriglyceridemia and pancreatic injury. Finally, Doxycycline, lisinopril, estrogens and steroids are associated with AP development, and constituted the chronic medications of some of COVID-19 patients.

  • Also, considering the high prevalence of hyperamylasemia/ hyperlipasemia among COVID-19 patients, but the low prevalence of AP in these patients, the manuscript would increase in value if the authors provided an algorithm/decision tree on how to approach increase in pancreatic serum enzymes in COVID-19 patients.

Response: Dear reviewer, since the scientific evidences about the topic are scarce and mainly heterogeneous, and due to the fact that our work represents simply a literature review, we prefer do not to propose an algorithm in order to not confound the reader about its real clinical usefulness and applicability

  • Besides the impact on pancreatic surgery, the authors could discuss the impact on pancreatic endoscopy also.  

Response: Dear reviewer, we added a paragraph about the impact of COVID-19 on pancreatic endoscopy also.

  • There is a small error on page 1, Line 38, “565 infections and 6 million deaths” – missing “million” for the no of infections.

Response: dear Reviewer, we corrected the mistakes accordingly

Many thanks again

Sincerely Yours

Emanuele Sinagra

This manuscript is a resubmission of an earlier submission. The following is a list of the peer review reports and author responses from that submission.